# Contagious Bovine Pleuropneumonia: A Passage to India

**DOI:** 10.3390/ani13132151

**Published:** 2023-06-29

**Authors:** Robin A. J. Nicholas

**Affiliations:** The Oaks, Nutshell Lane, Upper Hale, Farnham, Surrey GU9 0HG, UK; robin.a.j.nicholas@gmail.com; Tel.: +44-12-5272-5557

**Keywords:** CBPP, cattle, Assam, eradication, vaccines

## Abstract

**Simple Summary:**

With the eradication of rinderpest in 1995, contagious bovine pleuropneumonia (CBPP), the other great historical plague of cattle, has become arguably the most important bovine disease in sub-Saharan Africa affecting cattle in over 25 countries. CBPP, caused by the small bacterium *Mycoplasma mycoides* subsp. *mycoides*, is a contagious disease of cattle mainly affecting the lungs, leading to high morbidity and mortality. During the 19th and 20th centuries, it affected all cattle-rearing continents, but unlike its introduction into the USA, Africa and Australia from Europe, its origins in Asia are far from clear. This review examines the potential routes that cattle affected with CBPP may have taken, most probably from Australia into China and ultimately into the Indian province of Assam where outbreaks in the mid-20th century were unambiguously reported. An examination of reports contemporary to the outbreaks together with recent molecular analysis suggests that CBPP was introduced to India with affected cattle and buffalo from neighbouring countries, most likely China which was severely infected during this period. Evidence is discussed that CBPP may continue to plague cattle in parts of Asia.

**Abstract:**

The World Organization for Animal Health (WOAH)-listed contagious bovine pleuropneumonia (CBPP) emerged first in Europe and then spread to Eastern Asia, including Japan and China, from the Northern Territories of Australia at the end of the 19th century. Its route to India, however, is less well known as there is little evidence for large importations of cattle from Australia. The lack of accurate diagnostic tests at this time meant veterinary authorities relied solely on clinical and pathological signs, many of which were non-specific. Consequently, any diagnoses of CBPP reported in the early 20th century must be viewed with caution. More convincing reports of CBPP confirmed by laboratory tests were made in the 1930s and 1940s in the Indian state of Assam. Eradication campaigns began in the 1940s with immunizations of live attenuated vaccines and then more comprehensively in the 1950s and 1960s, supplemented with serological screening and the establishment of quarantine centres at international borders. The last case of CBPP, reported to WOAH, was seen in 1990, but the launch of a new awareness campaign in Assam in 2002 and recent reports of the disease in Pakistan suggests the disease has persisted in the Indian subcontinent well into the 21st century.

## 1. Introduction

Contagious bovine pleuropneumonia (CBPP), commonly known as “lung sick”, once a scourge of cattle worldwide, is now restricted to about 25 countries in sub-Saharan Africa with Ethiopia, Ghana, Tanzania and Angola having the highest prevalence [1]. Very recently, though unofficial, reports have indicated it may also be present in the Punjab region of Pakistan. The true economic and social impact of CBPP is unknown because rigorous surveillance is not routinely carried out in all African countries. Today, for the most part, CBPP is an enzootic disease with occasional serious outbreaks but it still takes a toll on cattle, resulting in losses due to poor growth rates, fertility, emaciation and a shortage of draft animals [2].

The characteristics of CBPP and the history of its worldwide spread has been well documented [2,3,4,5]. While it is not possible to completely exclude a CBPP-like disease in classical Roman times, it seems highly speculative given the lack of laboratory diagnosis and knowledge of disease pathology at the time. Descriptions of CBPP-like clinical signs were reported by Virgil but seems to have had a much wider host range, including dogs and birds, than seen today [3]. Furthermore, his descriptions of cattle vomiting blood and froth does not seem much like CBPP. The consensus, backed by molecular studies [5], is that this disease, known then as “pulmona,” first appeared in Middle Europe in the 16th century and was differentiated from other respiratory diseases by Bourgelat in 1765 [2]. It spread by wars and trade during the early 19th century throughout the continent, predominantly by movements of the favoured Swiss and Dutch cattle. The Netherlands became a hot bed of infection with high mortality and morbidity rates. Britain too became infected through the importation of cattle from mainland Europe, resulting in huge losses. CBPP was then exported from Britain and the Netherlands via infected cattle to the USA/Australia and South Africa, respectively. From the latter, CBPP-affected cattle were trekked north by Boer farmers [6] and/or British troops [7], initially into East Africa and then via other cattle movements into the rest of Africa. One hypothesis says that CBPP may have been introduced even earlier to Africa also by the British during a military expedition in 1868 into Abyssinia, now present-day Ethiopia [2]. Another theory suggests that it may already have been present in Africa, as the European explorer Thompson reported Masai cattle dying of a disease with signs similar to CBPP [6]. Though not mutually exclusive, Dupuy et al. [5] ingeniously suggested another possible route of transmission: via cattle from the Portuguese bases along the west African coast, possibly as early as the 17th century. However, the evidence for this speculation comes from molecular typing of the causative agent, *Mycoplasma mycoides* subsp. *mycoides*, which showed three separate genotypes in central, eastern and western Africa [8]. Unfortunately, the number of strains available was too few to provide a truly accurate assessment of the epidemiology of CBPP in Africa.

Various control measures including slaughter, quarantine, movement restrictions and vaccination led eventually to its eradication first in the USA in 1892, then Britain at the end of the 19th century, Australia in the 1960s, China in the 1980s and the whole of Europe by the end of the 20th century. Other countries, including India and Pakistan, reported their last cases in 1990 and 1997, respectively [9]. However, as will be seen later in this review, doubts exist about the complete disappearance of CBPP from the Indian subcontinent.

## 2. Asia

While the route that CBPP took to the USA, Africa and Australia is well documented, how and when it entered Asia is not so well known. The possibility that it entered China from Russia was partly discounted by multilocus sequence analysis, which showed that Chinese strains of *M. m. mycoides* were identical to the African/Australian cluster [10]. Australia began shipping cattle to Hong Kong and Singapore in 1884 from the Northern Territories, but trade was suspended because of disease outbreaks, most probably CBPP, five years later, with suspensions occurring sporadically after that. This strongly suggests that CBPP may well have entered Asia at this time. Salmon [11], writing about the origins of CBPP at the end of the 19th century, stated rather vaguely “*as existing in various parts of the continent of Asia though the time of its first appearance and extent of its distribution are very uncertain*…” In another contemporaneous account not long after CBPP had first reached Asia, Hutyra and Marek [12] declared that it was widespread in Mongolia and Manchuria (now present-day China and parts of the Russian Far East) as well as India and China, though what these claims were based on other than word of mouth is also uncertain. Seventy years later, ter Laak [13], using secondary sources, stated that New Zealand, mainland China, Mongolia, Myanmar, Bhutan, Nepal, Japan and India in the late 19th and early 20th centuries had also been infected from Australia. More authoritatively, in 1991, Lefèvre [14] in his “*Atlas des maladies infectieuses des ruminants*” located the disease in a swathe of Asia running from Sinkiang, in the west of China, south-eastwards towards Thailand and Vietnam covering Mongolia, Tibet, Bangladesh, Sichuan, Bhutan, Myanmar, Cambodia, Kampuchea and Assam; CBPP was also suspected in Pakistan and Nepal. However, despite the information deriving from the internationally renowned French Veterinary Institute in Alfort, Paris, there still must be doubt about the validity of some of the data, as they would have often been based on unreliable clinical reports from Asian countries without diagnostic facilities.

## 3. India

The exact date of arrival of CBPP in India has been hotly disputed and varies from the late 19th century to the early 1940s. A perusal of the Provincial Civil Veterinary Department’s annual reports by Gopalakrishnan [15] indicated that the opinion up to the late 1930s was always divided as to the existence of the disease in India. The earlier proposal by Windsor that the horn of Africa was infected during a British military operation which began in India is clearly unlikely, as CBPP had almost certainly not reached India at the time of the expedition in 1868. Furthermore donkeys, not bullocks, were the main beasts of burden for the invasion of Abyssinia [16]. Furthermore, there does not seem to have been any reports of major imports of cattle from Australia around this time, as trade did not begin for another 20 years [17]. Indeed, had there been any importation of infected cattle to India in the late 19th and early 20th centuries then there would almost certainly have been reports of explosive and prolonged outbreaks in the immunologically naïve local herds such as that seen in Australia and Southern Africa [4]. 

This is not to say that there were not major outbreaks of respiratory diseases in Indian cattle during the late period of British colonization. From 1892–1893, CBPP was included by the veterinary authorities in an annual list of diseases believed to occur in India, although any diagnosis at that time would have been based solely on clinical and pathological signs. In the following decade, thousands of cattle died with lesions thought to be CBPP. From 1894–1895, the superintendent of South Punjab reported an outbreak of “pleuropneumonia” in the Ferozepore area, comprising 138,000 cattle of which 4000 died. Similar diseases were recorded in the remarkably detailed annual reports of the Civil Veterinary Department between 1893 and 1904 in the same province, affecting 140,636 animals of which 5631 died. However, there were doubts expressed by some authorities at the time, including those in Bengal, that this was not true CBPP and, with hindsight, was most likely bovine haemorrhagic septicaemia caused by *Pasteurella multocida*. Indeed, the veterinary superintendent at the time stated the following: “*It was much more likely that these were ordinary pneumonias with pleurisy*” [18].

Identification of diseases, particularly multifactorial respiratory conditions in cattle in the late 19th century, was not an exact science, as seen with a transatlantic trade dispute between the USA and Great Britain. An argument had broken out over the on-going diagnoses “…*of the insidious lung disease pleuropneumonia*” in cattle imported from the USA in 1879 [19]. Despite the irony of Britain having exported CBPP to the USA in the early half of the 19th century and at the time being heavily affected itself, British veterinary officials wanted to slaughter cattle imported from the USA that were diagnosed as being affected with CBPP. However, these diagnoses were contested by both the US authorities and some experts in Britain who believed it was a general non-contagious and less fatal bronchitis, known today as “shipping fever”, which had developed during a stressful transatlantic voyage. While never satisfactorily resolved, the dispute accelerated US attempts to eradicate CBPP, which it did in 1892. Britain suffered for a few years more.

In India, in the early part of the 20th century, sporadic reports of a CBPP-like disease were made near Karachi, the United Provinces, Berar, Jubbulpore and Nagpor districts with high mortality, although once again, CBPP could not confirmed. One convincing reason that these diseases may have been misdiagnosed was that there were few further reports of CBPP between 1905 and 1929, but instead, a new category was inserted into the records, namely haemorrhagic septicaemia, suggesting doubts in the earlier diagnoses. As is well known, CBPP does not disappear so suddenly following 10 years of infection without severe control measures being put in place for which there was no evidence in India. The reasons for these doubts must have been due to staff inexperienced at differentiating the various bovine pneumonias and the uncertainty of disease nomenclature at the time, compounded by the huge distances and difficulties encountered by veterinarians examining diseased cattle in the field.

Further arguments for the absence of CBPP up to that time came from two senior veterinary officers serving in India in the early 20th century. J.T.Edwards in 1927 said that “…*India…for some reason has escaped infection*” [18]; 12 years later, Shirlaw [18] stated, in no uncertain terms after viewing reports dating back to the beginning of that century, that “*There is no evidence that CBPP existed or has ever existed in British India*”. While Shirlaw was probably correct about the historic reports, he believed that many of the later reports of a CBPP-like disease in Assam, an isolated state in northeast India bordered by Bangladesh, Myanmar, Bhutan and China (Figure 1), were caused, based on culture and histopathology, by a fungal pathogen. The mycotic-looking *Borrelomyces peripneumoniae*, later to be known as the bacterium *Asterococcus mycoides* then renamed *Mycoplasma mycoides*, the definitive causative agent of CBPP, was later identified from the pleural fluid of affected cattle. Five years after his pronouncement of its non-occurrence in India, Shirlaw had to admit that his fungal disease was in fact true CBPP and ravaging cattle in Assam [20]. He and his co-workers at the Imperial Veterinary Research Institute, Izatnagar, Uttar Pradesh were able to develop “a medium of choice” for the in vitro culture of the “Assam bovine pleuropneumonia organism” and showed that strains from this province were immunologically similar to confirmed CBPP pathogens from East Africa and Australia. It is interesting to note that despite taking 3 and 10 weeks to arrive from Kenya and Australia, respectively, the strains could be revived satisfactorily in the newly developed broth and still retained pathogenicity in cattle.

## 4. Assam

It is highly likely that the first true cases of CBPP in Assam were the deaths of 30 cattle reported between 1932–1933. Clear unambiguous descriptions were given of the pathology with all salient features of the disease known locally as “*Brahmaputra Valley Disease*” were seen, although the mycoplasma was not isolated or detected, owing to the lack of culture facilities [20]. From then on, reports of CBPP were made annually. Laboratory investigations began seriously at the Imperial Veterinary Research Institute around 1936, with the isolation of the pathogen from pleural fluids and pieces of affected lungs from cattle in Assam. Experimental subcutaneous inoculation of disease-free cattle with pleural fluids from infected animals led to large swellings at the inoculation site and, less frequently, CBPP-like lung lesions as seen in the field; this highlighted the difficulty in reproducing the clinical disease. Better results were seen when the cattle inhaled or were intubated with large quantities of the mycoplasma, though this was often inconsistent [21]. While intubation of mycoplasma direct into the bronchioles is often the most common method these days, contact infection of healthy cattle by animals incubating the disease is probably the best means of inducing natural infection [22].

In March 1938, the disease was reported as enzootic in certain areas surrounding the Brahmaputra River Valley. It was believed to be seasonal and was observed more frequently during the monsoon period. In acute cases, the body temperature reached 40–41 °C with both lungs showing lesions. However, in chronic cases, only one lung was usually involved. Post-mortem findings showed the lungs of infected cattle were hepatized and adhered to the chest wall which contained large amounts of sanguineous pleural fluids [23]. All features described were classical CBPP lesions as seen today in sub-Saharan Africa. The disease was particularly prevalent in low-lying areas near the riverbank where many herds grazed. It was reported as spreading to domestic buffalo, but this species was much less susceptible and showed few clinical signs. The isolated nature of the area had two conflicting outcomes: first, control was clearly hampered by the remoteness of the region and secondly and fortunately, apart from a single outbreak in the Sylhet district (in present-day Bangladesh), there was no detectable spread of the disease to the rest of India [21]. Other factors affecting the pace of control were the long incubation period of the disease, which was reported to vary between 1–8 months, and the insidious and silent spread of the disease through the herd [18]. It was also believed that the high humidity of the region enabled the enhanced survivability of the normally fragile pathogen [21]. 

The number of cattle with CBPP reached a peak in 1956, with 1471 affected and 1036 dying. By 1963, only 128 were affected of which 10 had died (Table 1). After this, cases became more sporadic, and by 1965, only three districts remained affected. This turnaround was due largely to the introduction of a vaccine made from a local strain, the pathogen, which had been attenuated by serial passage in a broth and used as a vaccine following 12–25 subcultures. The vaccine was injected into the tail tip which often caused necrosis, requiring treatment with the anti-inflammatory medication, sulphamezine. Despite this, the tail-tip site was adopted as the optimum route, because few lung lesions resulted following experimental challenges with immunity lasting about one year. Vaccine records compiled by the veterinary authorities showed the numbers of animals vaccinated varied from 20,000 at the beginning of the 1950s to 86,000 towards the end of the decade, with a gradual levelling off in 1962 as the disease came under control (Table 1).

## 5. Origins

The origins of CBPP in India have been much debated. Rao [24] speculated that the disease may have existed in Assam as early as 1919, while others thought it might have been present at the beginning of the 20th century [11,12]. Singh and Rana [23], writing nearly 70 years later, cautiously date the first confirmed case of CBPP as late as 1942. This was based largely on the introduction of better confirmatory laboratory tests, in particular the agglutination test, and a pathognomonic description of lesions recorded by veterinarians of the day. Das [21] also put the date of the first detection as late as 1944. However, at this time, CBPP was already endemic in the Naga Hills and Manipur, affecting at least eight districts containing, in all, 100,000 cattle belonging to about 1000 villages [15]. This uncertainty is perhaps not that surprising given that the Japanese Imperial Army were attempting to invade India from Myanmar at this time and the fact that any animal health surveillance being conducted close by was quite remarkable. Clearly, the exact date of entry will never be resolved, but it is difficult to know where CBPP would have originated prior to Australian exports to Asia considering the isolated nature of the region. A more likely explanation is that CBPP entered Assam via affected cattle and yaks in the 1930s and 1940s from the neighbouring countries of China, Bhutan and/or Myanmar where the disease was believed to have existed and its spread facilitated by the lack of border checkpoints. More precisely, the occurrence of the disease in the riverain tracts of Lakhimpur, which is located in the northeast corner of Assam, suggests the source of infection was most likely to be China. Indeed, it was well known that much of China, including provinces close to India, was affected by CBPP following its introduction into Shanghai from Australia [25]; indeed, only 6 of the 23 provinces remained free of the disease between 1949 to 1969 when nearly 200,000 animals were reported to have died of CBPP, putting a great strain on the economy during the “Great Leap Forward”, which relied heavily on cattle for labour, meat and milk [25].

To control animal movements, quarantine stations were established on the Bhutan border in the 1950s at the start of the eradication campaign; the health status of cattle were then checked by veterinarians, which included the use of the rapid agglutination tests for CBPP. No records have been found showing whether any positive animals were detected at these stations. While the last official case of CBPP was reported in Assam in 1990, a new eradication programme, which was a component of the rinderpest eradication project, was reported by a provincial newspaper Sentinel based in Guwahati [26]. The programme was launched in March 2002 by M. K. Barooah, Assam’s Commissioner and Secretary of the Veterinary Department, strongly suggesting that CBPP was continuing to cause problems, and greater awareness of the disease by cattle farmers was deemed necessary. No more information could be found on the outcome of the awareness campaign, so it must be assumed no further cases of CBPP were reported to the authorities.

## 6. Freedom from CBPP

Three years after the last reported clinical case in 1987, India stopped vaccinating cattle for CBPP [23]. Then, to show provisional freedom from the disease decreed by the WOAH, a programme was launched in 2001 by the Department of Animal Husbandry and Dairying and Ministry of Agriculture, New Delhi in collaboration with the Indian Veterinary Research Institute, Izatnagar and the Department of Veterinary and Animal Husbandry, Government of Assam [23]. Bovine tissue samples were collected and tested for both the pathogen and its antibodies; this was undertaken jointly by the Mycoplasma Laboratory, IVRI, Izatnagar and by the CBPP laboratory, Khanapara. No CBPP cases were detected, and India was declared provisionally free from the disease, taking effect from October 2003. Seven years later, on 26 May 2007, the WOAH declared India free from CBPP infection, as per OIE resolution number 17 (General Session 82 of May 2014). Other countries comprising Argentina, Australia, Botswana, Canada, China, Portugal, Singapore, Switzerland and the USA also obtained CBPP-freedom status at that time. 

## 7. Pakistan

It is tempting to speculate that the origins of CBPP in Pakistan must be tied very closely to that of India as the countries only separated in 1947. However, while there have been no confirmed reports of CBPP in India, except in Assam, since Partition, it looks to have continued to plague Pakistan as evidenced by recent reports by Anjum et al. [27,28]. To date, however, the extent of the disease in the country is not known. A recent series of clinical and seroprevalence studies have reported the presence of CBPP in several districts of the Punjab province, near the border with India [27,28,29], but official notifications have not, at the time of writing (June 2023), been made to the EFSA Panel on Animal Health and Welfare or the WOAH. In these studies, lung samples from over 500 cattle and nearly 300 buffalo, suspected clinically of having CBPP, were collected from abattoirs in three districts of Punjab, namely, Lahore, Kasur and Jhang. The results indicated that just over 9% of cattle were positive by a PCR detecting the 16S rRNA gene; all buffalo samples were negative for CBPP. These results were very similar to earlier studies which also showed a seroprevalence of 10% [27]. This is surprising perhaps because antibodies are usually more widespread in herds than the pathogen itself. It should also be pointed out that the PCR used was not particularly specific, as members of the *M. mycoides* cluster cross-react; however, the only other large ruminant member of the cluster is *M. leachii*, a rather rare organism more associated with mastitis and arthritis. Further evidence that the authors detected *M. m. mycoides* is that the sequenced isolates from Pakistan grouped phylogenetically with Chinese isolates and less so with Australian and an Italian isolate but are clearly separate from other members of the cluster [28]. Unfortunately, no pathology of the affected cattle has been described to show whether lesions are acute or chronic. Incidentally, higher prevalence of the disease was seen in cattle of more than seven years of age, in female cattle and in cross-bred cattle. Whether India is at risk from these affected Pakistani states is not clear but should warrant increased surveillance near the border.

## 8. Conclusions

There are several possibilities for the origin of CBPP in India: its direct arrival in cattle from Europe, most probably Britain; its direct importation from Australia; or its introduction from the neighbouring countries of China, Bhutan or Myanmar, which would have also had origins in Australia. A final possibility is its importation through China from Russia which was affected in the 19th and early 20th centuries, though the geopolitics suggests this is the least likely route [25]. Experience from Africa showed that without attempts to stamp out CBPP immediately on arrival in South Africa from the Netherlands, it spread progressively north, marked by large outbreaks with high mortality when infected cattle came in contact with immunologically naïve local breeds. It has been reported that over 100,000 cattle died within two years of its introduction [4]. Additionally, in Australia, the lack of control when the disease was first identified led to a century of CBPP characterized by large and episodic outbreaks. There appears to be no similar descriptions of prolonged CBPP-like diseases in India in the late 19th and early 20th centuries. While there were reports of a respiratory cattle disease with high mortality at this time, it seems very likely that these reported CBPP outbreaks were misdiagnosed and were, in fact, haemorrhagic septicaemia caused by *Pasteurella multocida*, which does not persist so stubbornly. Indeed, some experts at the time, including J F Shirlaw, were convinced that CBPP had never infected India until the confirmed cases in Assam in the late 1930s and early 1940s [18]. Due to the isolated nature of this Indian province and its lack of trade with the rest of India, it is highly likely that CBPP was introduced with cattle and buffalo from China, or the neighbouring countries mentioned earlier. 

The presence of CBPP in Pakistan is less well known, but recent reports indicate that the clinical disease is rife in Punjab [29], an historical and politically sensitive area shared with India, though separated by an intensely monitored 3000 km border fence with only a single crossing point. Consequently, the risk to India of transboundary diseases such as CBPP from Pakistan that require cattle movements to spread is low, unless trade restrictions between the two countries change. Accepting the likelihood that CBPP is unknown to the main landmass of India, the origins in Pakistan must lie elsewhere, most probably in China, as studies by Anjum et al. [28] have shown. According to the WOAH, the last case officially reported in Pakistan was 1997, which suggests that the disease has persisted in cattle since then but has gone unreported. Indeed, it is well known that CBPP can exist in small herds silently, with cattle exhibiting few clinical signs as it probably did in Europe in the late 20th century [16]. While CBPP has only been reported to date in the state of Punjab, which constitutes the main cattle farming region of Pakistan, those areas such as Khyber Pakhtunkhwa and Balochistan, which are closer to the Chinese border, may also be affected. Suspicion must also be cast on the occurrence of CBPP in adjacent countries such as Bhutan and Myanmar, where little surveillance is carried out and borders are less secure. Interestingly, a nationwide study of cattle in Pakistan’s close neighbour, Afghanistan, has showed no serological evidence of CBPP [30]. While the survey covered only 0.03% of the Afghan cattle population, it would still be capable of detecting CBPP at a prevalence of at least 1%. Whether the collapse of veterinary services following the invasion by the Taliban in 2021 will affect this disease-free status remains to be seen.

## Figures and Tables

**Figure 1 animals-13-02151-f001:**
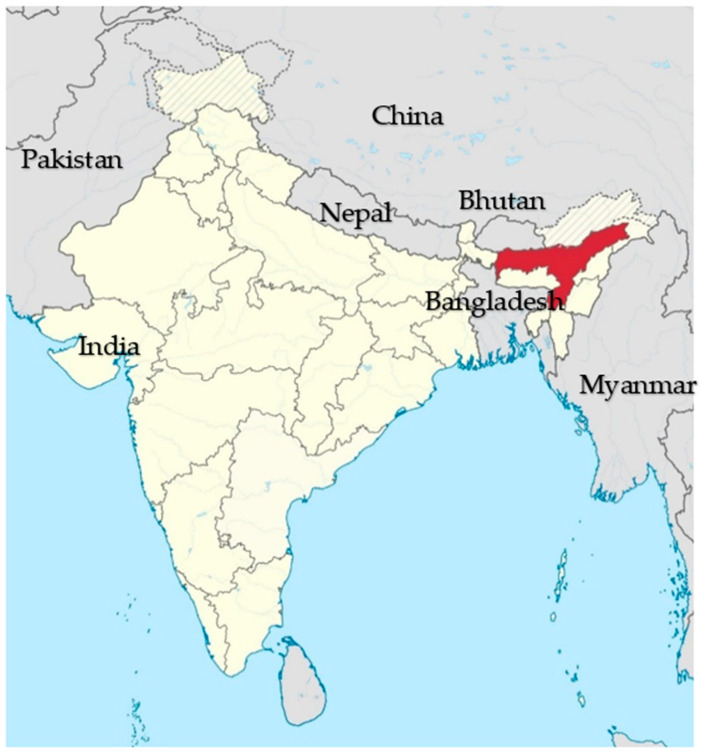
Location of the Indian state of Assam (in red) with its neighbours.

**Table 1 animals-13-02151-t001:** CBPP cases in cattle and vaccines delivered in Assam, 1951–1963 [Rao, 1969].

Year	Number Affected	Number Died	Vaccine Doses Given
1951	168	134	0
1952	621	604	0
1953	681	601	20,000
1954	152	137	20,000
1955	1065	662	25,000
1956	1471	1036	21,000
1957	745	260	43,000
1958	221	125	66,000
1959	55	32	86,000
1960	117	49	64,000
1961	293	27	74,000
1962	185	25	70,000
1963	128	10	52,000

## Data Availability

The scientific reports and websites consulted are all publicly available or can be obtained from the author.

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
