# Peer review of "Contagious Bovine Pleuropneumonia: A Passage to India"

_animals, 2023, doi:10.3390/ani13132151_

Round 1

Reviewer 1 Report

The paper provide an interesting overview on the history of CBPP occurence in India and Asia highlighting facts and gaps of historical knowledge of disease introduction, spreading and eradication.

The review is very well structured and easy to read.

Minor comments.

Line 232: I would add the word "animals" after "200,000" (200,000 animals...)

Line 236: I would add the word "status" after "health" (The health status of...)

Line 253: remove fullstop after "Government"

Line 269-270: Double check the sentence ".....the EFSA Panel on Animal Health and Welfare at WOAH). The panel on Animal Health and Welfare belong to EFSA and not to WOAH.

Line 279: Add "to" after associated (....more associated "to" mastitis and arthritis)

Author Response

The paper provide an interesting overview on the history of CBPP occurence in India and Asia highlighting facts and gaps of historical knowledge of disease introduction, spreading and eradication.

The review is very well structured and easy to read. I thank the reviewer for his comments

Minor comments.

Line 232: I would add the word "animals" after "200,000" (200,000 animals...)     Done

Line 236: I would add the word "status" after "health" (The health status of...)       Done

Line 253: remove fullstop after "Government"                                                          Done

Line 269-270: Double check the sentence ".....the EFSA Panel on Animal Health and Welfare at WOAH). The panel on Animal Health and Welfare belong to EFSA and not to WOAH. The reviewer is correct and I have modified the sentence

Line 279: Add "to" after associated (....more associated "to" mastitis and arthritis)    Done

Reviewer 2 Report

Dear Dr. Nicholas, 

This manuscript represents an excellent example of research in the area of veterinary history, in a well documented and interesting manuscript aimed both at a general audience and to research in the field of mycoplasmology and cattle diseases, by shedding some light regarding a neglected cattle disease of extremely high relevance for many countries, both currently and in the past.

Some minor comments can be found below:

The author should be consistent with use of the abbreviation of the causative agent's name: example in the text include the full name Mycoplasma mycoides subsp. Mycoides (55-56), M. m... mycoides (line 69), Mycoplasma m.mycoides (line 152), M. m. mycoides (line 280). Please unify criteria.

Line 178: Modify punctuation: infection [22].

212, 234, 326: change () for [] in the references

247: remove : in the paragraph title

278: Add . after M

Reference 17 should be checked as it does not seem to be working

Author Response

 Thanks for positive comments

The author should be consistent with use of the abbreviation of the causative agent's name: example in the text include the full name Mycoplasma mycoides subsp. Mycoides (55-56), M. m... mycoides (line 69), Mycoplasma m.mycoides (line 152), M. m. mycoides (line 280). Please unify criteria.   

I hope I have satisfied the reviewer but I have corrected the name at L152 now L165 because the existence of the LC type was not known at that stage so was just known as Mycoplasma mycoides. I have left the full genus name because of the existence of the two earlier incorrect genus names

 Line 178: Modify punctuation: infection [22]. Done

212, 234, 326: change () for [] in the references Done

247: remove : in the paragraph title Done

278: Add . after M   Done

Reference 17 should be checked as it does not seem to be working

I have changed this reference to the correct one

Reviewer 3 Report

While the story of CBPP spread is interesting and mysterious, the manuscript does not significantly contribute much to the field. The discussion of CBPP spread is not well organized and not enough background information is provided on why CBPP is an important disease. 

Additional comments:
- The manuscript discusses potential routes of CBPP spread into India.
- The manuscript discusses an original topic and provides a summary of potential routes of spread for CBPP into India. However, the manuscript does not emphasize the importance of CBPP spread and why the readers should be interested in this topic.
- The main question addressed by the research: The authors discuss alternative means of CBPP introduction and spread to India
- Authors should consider reorganizing the manuscript to provide a stronger story on CBPP in India. The current organization can leave the readers confused to why certain countries and their CBPP cases were discussed.
- The conclusion does a nice job of summarizing CBPP spread and describes various theories of CBPP spread to India. However, the manuscript leaves the readers to think why is it important to know about CBPP spread in India? What is the big picture/impact for this manuscript?
- The references are appropriate.

Author Response

While the story of CBPP spread is interesting and mysterious, the manuscript does not significantly contribute much to the field. 

I am a bit surprised by this comment as there have been no papers published on CBPP in India for over 70 years apart from brief and vague references to its existence in reviews. I have used original references dating back to the 1930s and even earlier that have not seen the light of day for many years. I think the discussion as to whether CBPP existed or not is 'novel, very interesting and mysterious' (as reviewer states) particularly its possible misdiagnosis at the turn of the 20th century

The discussion of CBPP spread is not well organized and not enough background information is provided on why CBPP is an important disease. 

I believe the organisation is logical: it begins with CBPP in the wider world then focuses on Asia, India then Assam where the confirmed outbreaks were seen. I then discuss the possibility of its origin there and the possible continued existence in those neighbouring countries. The existence of CBPP in Pakistan is relatively new finding which has not been notified officially yet.  As to more background information on CBPP, I believe I have covered the main characteristics and referred to major reviews that cover it more appropriately. L45-46. I have also mentioned its severe impact in European countries, South Africa etc throughout. The main focus was on its emergence in Asia which I think will be of interest to many authorities on this well known disease

Additional comments:
- The manuscript discusses potential routes of CBPP spread into India.  Yes
- The manuscript discusses an original topic and provides a summary of potential routes of spread for CBPP into India. Yes

However, the manuscript does not emphasize the importance of CBPP spread and why the readers should be interested in this topic. 

I would humbly beg to differ here, a view shared by the two other referees

  • - The main question addressed by the research: The authors discuss alternative means of CBPP introduction and spread to India  yes
    - Authors should consider reorganizing the manuscript to provide a stronger story on CBPP in India. The current organization can leave the readers confused to why certain countries and their CBPP cases were discussed.  See my comments above 
    - The conclusion does a nice job of summarizing CBPP spread and describes various theories of CBPP spread to India. Thanks
  • However, the manuscript leaves the readers to think why is it important to know about CBPP spread in India? What is the big picture/impact for this manuscript?  
  • I appreciate the comments but feel much of my response to the criticism is self evident. I believe authorities in the sub continent will be interested as will historians of science and those who have seen the disease in Africa and till quite recently Europe. I hope it will refocus attention to CBPP where its presence is still suspected

Round 2

Reviewer 3 Report

The addition of the simple summary is excellent and nicely summarizes the objective of the manuscript and why CBPP is important regarding animal health. This summary will help guide the reviewer through the historical review of CBPP in Asia.

The author responded to all the reviewer comments and provided strong justification for organization and purpose of the manuscript. This is a thorough review of the literature relating to CBPP movement globally and is intriguing from a historical point of view.